# Ashi Scalp Acupuncture in the Treatment of Secondary Trigeminal Neuralgia Induced by Multiple Sclerosis: A Case Report

**DOI:** 10.3390/medicines9080044

**Published:** 2022-08-12

**Authors:** Qiong Schürer, Hamdy Shaban, Andreas R. Gantenbein, Giada Todeschini, Saroj K. Pradhan

**Affiliations:** 1Research Department, Swiss TCM UNI, 5330 Bad Zurzach, Switzerland; 2Department of Psychiatry Private Clinic of UPK, The University Psychiatric Clinics, 4002 Basel, Switzerland; 3Neurology & Neurorehabilitation Department Rehaklinik, ZURZACH Care, 5300 Bad Zurzach, Switzerland; 4Research Department Rehaklinik Bad Zurzach, TCM Ming Dao Bad Zurzach/ZURZACH Care, 5330 Bad Zurzach, Switzerland

**Keywords:** secondary trigeminal neuralgia, Ashi point, scalp acupuncture, multiple sclerosis

## Abstract

**Background**: Multiple sclerosis (MS) is an autoimmune, chronic, inflammatory, demyelinating, and axonal degeneration disease of the central nervous system. Trigeminal neuralgia (TN), a neuropathic facial paroxysmal pain, is prevalent among MS patients. Because of the inadequacy of the comprehension of MS-related TN pathophysiological mechanisms, TN remains arduous in its treatment approaches. Acupuncture as a non-pharmacological therapy could be a promising complementary therapy for the treatment of TN. MS gradual neural damage might affect the muscles’ function. This can lead to acute or paroxysmal pain in the form of spasms that might progress to formation of myofascial trigger points also known in traditional Chinese medicine as Ashi points (AP). Localising these AP through palpation and pain sensation feedback in patients with MS is an indicator of disease progression. Pathologically, these points reveal the disharmony of soft tissue and internal organs. Methods: This case report examined the pain relief outcome with Ashi scalp acupuncture (ASA) in a secondary TN patient who was unsuccessfully treated multiple times with body acupuncture. The main outline measure was to quantify pain intensity using a numerical rating scale (NRS) before and after each acupuncture therapy. The patient was treated on the scalp for a total of eight times, twice a week over four weeks. **Results**: A reduction in secondary TN pain intensity was observed after each session. On average, the patient expressed severe pain (NRS: 8.0 ± 2.20) before ASA treatment, which significantly decreased after therapy to mild pain (NRS: 2.0 ± 1.64). **Conclusions**: Significant improvements in pain intensity reduction after each acupuncture treatment without any adverse effects were observed.

## 1. Introduction

Multiple sclerosis (MS) is an autoimmune-mediated neurological disorder that affects the central nervous system (CNS), leading to chronic illness affecting the brain and spinal cord. The underlying mechanism of an MS attack is an aggressive inflammation process in areas of the white matter of the central nervous system in random patches called plaques. Upon formation of theses plaques, the destruction of myelin sheaths is triggered. Myelin is the insulating sheath of nerves that promotes smooth and high-speed transmission of electrochemical messages between the brain, the spinal cord, and the rest of the body. The damage of the myelin sheath leads to disruptive neuronal communication and slowed electrochemical signalling. The blockade of neuronal transmission causes loss of function and somatosensory disturbance [1].

Trigeminal neuralgia (TN) involves one or more branches of the trigeminal nerve. Orofacial pain is among the most distinctive and difficult to cure in patients with MS with a prevalence of 0.1% to 9.7% [2]. TN is characterized by severe, mostly unilateral pain. The electric shock-like, stabbing, or sharp pain usually lasts only a few seconds, sometimes up to two minutes, and can be accompanied by contractions of the mimic muscles, reddening of the face, and secretion of tears and sweat [3]. The «*The International Classification of Orofacial Pain, 1st edition*» provides an extensive description of all categories of pain in the orofacial area with a specific benchmark for all of them [4].

The Western medicine pharmacological first-line treatment for TN is with carbamazepine and oxcarbazepine. Other treatment choices include lamotrigine, phenytoin, clonazepam, gabapentin, pregabalin, topiramate, levetiracetam, tocainide, and surgery [5].

Acupuncture as a non-pharmacotherapy seems more effective in the treatment of TN than pharmacotherapy or surgery [6]. 

In traditional Chinese medicine (TCM), TN is classified into three different categories. TN is caused by wind and cold, wind and heat, and evoked by qi and blood stagnation or congestion [7]. Ashi points (AP) were first specifically described in Sun Si Mao’s (AD 580-682) book «Prescriptions Worth a Thousand Pieces of Gold for Emergencies», but the idea originated from «The Yellow Emperor’s Inner Canon» [8]. AP is a point-locating method obtained through palpation on a certain site. AP are not always located in the area of pain [9] and are not associated with any meridians. Pathologically, these points reveal the disharmony of soft tissue and internal organs. 

The demand for acupuncture in pain management is steadily growing amongst patients [10]. Several clinical trials have shown the efficacy of acupuncture in the treatment of TN with satisfactory results [11,12]

Although the patient had previously and unsuccessfully undergone a total of 27 body acupuncture treatments, our approach was to use the scalp Ashi acupuncture points (AAP) on the scalp. We aimed to examine whether the Ashi scalp acupuncture (ASA) method could contribute to the pain relief management immediately after each acupuncture session and over a certain period. Our clinical experience over a decade has shown efficacy in pain relief when ASA was performed for head and facial region symptoms. 

The study was conducted according to the criteria set by the Declaration of Helsinki and written informed consent was obtained from the study participant before participating in the study. Ethical approval was not required for the single described case.

## 2. Case Description

A 46-year-old man diagnosed with secondary trigeminal neuralgia (TN) caused by MS was admitted to our rehabilitation centre (RC) as part of the inpatient, multimodal Zurzach Headache Program. He was diagnosed with MS in 2014. At that time, his main symptom was a recurrent pain (in the context of TN) in the right half of his face. As he had taken part in an in-patient program in another RC in 2016, treatment was implemented for his repeated psychosomatic state of exhaustion, which had also been an early symptom. In 2018, the patient mentioned a significant increase in attack frequency with typical TN pain. In the same year, cranial magnetic resonance imaging performed from an external hospital showed an MS plaque in the core area of the trigeminal nerve. 

Due to the patient’s anxiety to undergo surgery, he was treated as an outpatient 27 times with acupuncture over a three-month period. The following acupuncture points were punctured: ST 2, ST 7, BL 2, LI 4, ST 44, and LV 3. Each time during or after acupuncture, he experienced an adverse effect and the TN pain became more severe.

Two months later, he was treated with radiofrequency thermocoagulation of the Gasserian ganglion performed with intubation anaesthesia. This therapeutic approach brought the patient a minimal improvement in his condition. Five months later, after a test trial, direct nerve stimulation was implanted under his forehead and cheek. Thereafter, the patient was free of trigeminal pain for about six months. However, subsequently, the attack frequency and pain intensity re-occurred. In our RC, he reported experiencing facial unilateral, stabbing, sharp pain lasting a fraction of a second, but appearing repeatedly up to 100 times per day. Aggravating factors of pain were triggered by swallowing, speaking, and stress. He had been unable to figure out strategies to alleviate the pain to date.

Besides MS and secondary TN, the patients’ comorbidities included hyponatremia, metabolic syndrome, recurrent depressive episodes, cardiac disease, and non-alcoholic fatty liver disease. 

### 2.1. Treatment with Western Medicine 

The patient was already taking several medications before RC admission. Table 1 provides an overview of his medication list. Interferon beta-1a and Dimethyl fumarate were adjourned in 2014 because of adverse drug reactions. The patient then started a therapy with teriflunomide in 2018. He was therapy-resistant against pregabalin, oxcarbazepine, and fentanyl. However, he responded completely well to gabapentin in 2019.

### 2.2. Treatment with Ashi Scalp Acupuncture

At each therapy session, AAP on the scalp were searched bilaterally with an acupuncture pinhead and subsequently punctured. Each acupuncture session lasted 50–60 min, which is standard in our RC. Neither tonifying nor reducing manipulation needle technique was implemented. 

The acupuncture direction was an oblique angle. The puncture technique was <15° shown in Figure 1. The patient was lying in a supine position during the treatment. 

Sterile disposable 0.25 × 25 mm stainless-steel, single-use needles, manufactured by Asia-med GmbH (Pullach, Germany) were used. 

A licensed physician in TCM with over 10 years in clinical practice executed the therapeutic regime.

### 2.3. Progression of Symptoms

The patient received individualized ASA twice a week, for a total of eight sessions for four weeks at the RC.

The documentation was carried out by means of a pain questionnaire with a numerical rating scale (NRS, 0 = no pain to 10 = the worst imaginable pain) [13]. According to Salaffi et al., a reduction of 1 cm on the 0–10 cm numerical rating scale represents a “slight improvement in pain” (slightly better) for the patient and a reduction of 2 cm represents a “marked improvement in pain” for the patient. A reduction of 1 cm was defined by Salaffi et al. as the smallest clinically important difference in pain perception [14].

After each acupuncture session, the patient reported a significant reduction in pain intensity. Average NRS was 8.0 (sd = 2.20) before acupuncture and 2.13 (sd = 1.64) after acupuncture. This results in a mean difference of 5.88 (sd = 2.03), corresponding to a standardized response mean (Cohn’d) [15] of 3.07 (95%-CI: 4.14–7.66, Two-tailed *p* < 0.00009), meaning a large significant effect. For the first time within the past four months, the patient was pain-free for 36 h after the third acupuncture treatment. This pain-free state re-occurred after the seventh acupuncture treatment, which lasted 36 h. The patient was completely free of pain after the last acupuncture session until his discharge from our RC. Table 2 and Figure 2 and Figure 3 illustrate the results of pain intensity assessment using the NRS before and after acupuncture treatments by therapy session. 

A three- and six-weeks follow-up was carried out via telephone, in which the patient reported a pain-free status (NRS 0 = no pain). Changes in the NRS between discharge and follow-up were evaluated by standardized mean difference. Score changes on the NRS, from discharge time and follow-ups, remained unaltered (NRS 0 = no pain). 

The patient reported very positively at the last visit and at follow-ups. He mentioned that he was free of pain and his quality of life had improved significantly. Symptoms of depression were also no longer to be observed. 

No adverse effects were observed during all acupuncture sessions.

## 3. Discussion

In this case, a substantial short-term pain reduction was observed after all acupuncture session’s standardized response mean (SRM) (SRM = 3.07) with the ASA technique. The size of those effects may be clinically relevant because SRMs ≥ 0.50 are very likely to be subjectively perceived by the patients [16]. The hypothesis that the subjectively perceived secondary TN pain is less severe after the application of ASA technique than prior therapy has been confirmed in this case. ASA seemed to be superior in secondary TN pain management when compared to classic body acupuncture. 

ASA treatment was effective while the patient experienced adverse effects when he received meridian acupuncture (ST 2, ST 7, BL 2, LI 4, ST 44, and LV 3). ASA points in Figure 1 differ from GB 15, GB 16, GB 17, and GB 18. These latter mentioned points mainly treat paresis, headache, vertigo, cervical pain, nausea, Eyestrain, and fever. Therefore, it is important to emphasize that Ashi points reveal the disharmony of soft tissue and internal organs.

In literature, many reports about scalp acupuncture (SA) can be found. In their case report, Hao et al. (2013) concluded that SA was a better therapy approach in bringing about rapid advancement in patients with MS correlated to other acupuncture methods to the same degree as acupuncture on the ear, body, and hand [17]. In a randomized controlled clinical trial, the initial SA interference after acute ischaemic stroke with hemiplegic paralysis showed an enhanced myodynamia of the afflicted limbs, but also an improvement of neurological deficiency severity and quality of life [18]. A meta-analysis result reported that SA improved motor function in patients with stroke, whether ischemic or hemorrhagic [19]. In a systematic review and meta-analysis evaluating the efficacy and safety of SA for insomnia, the authors could provide insight of the effectiveness and safety treating insomnia [20]. Liu et al. (2019) conducted a systematic review and meta-analysis about SA in children with autism spectrum disorders (ASD). The main results in their meta-analysis advocated that in comparison with behavioral and educational interventions, SA was more effective in lowering Childhood Autism Rating Scale and Autism Behavior Checklist scores and promoting Psychoeducational Profile scores in communication, physical ability, and behavior. Accordingly, the authors ensured that SA may show positive effects on the treatment for children affected by ASD [21]. Therefore, patients with MS, as well as patients who have suffered from a stroke, patients with insomnia, and children with autism spectrum disorders might benefit from SA. In previous years, large efforts were made to understand the effect of acupuncture in MS [17,22,23] and TN [6,24]_,_ and further research studies are still ongoing. The outcome of these studies is favourable for acupuncture. 

These studies have been carried out in patients with a broad spectrum of neurological and non-neurological diseases, and research still has to be carried out in several fields of Western medicine, but the studies carried out until now have shown favourable results for the effectiveness of acupuncture as a symptom-reliever. 

Moreover, Edwards et al. (2020) showed that acupuncture was more effective than pharmacotherapy or surgery in the management of TN, cost-effective in comparison to surgery or pharmaceutics, and caused significantly fewer adverse effects [6].

The limitation of this study is that a follow-up of 3 and 6 months could not be conducted due to the inaccessibility of the participant during the pandemic. 

In conclusion, the outcome of this case report provides findings that ASA has potential in the treatment of trigeminal pain management. The local pain network modulation with Ashi acupuncture can possibly change pain perception in patients who were unsuccessfully treated with “general” acupuncture before obtaining a more specific and targeted treatment. We do recommend experts in acupuncture to make use of our method with patients who have been previously unsuccessfully treated with body acupuncture.

## Figures and Tables

**Figure 1 medicines-09-00044-f001:**
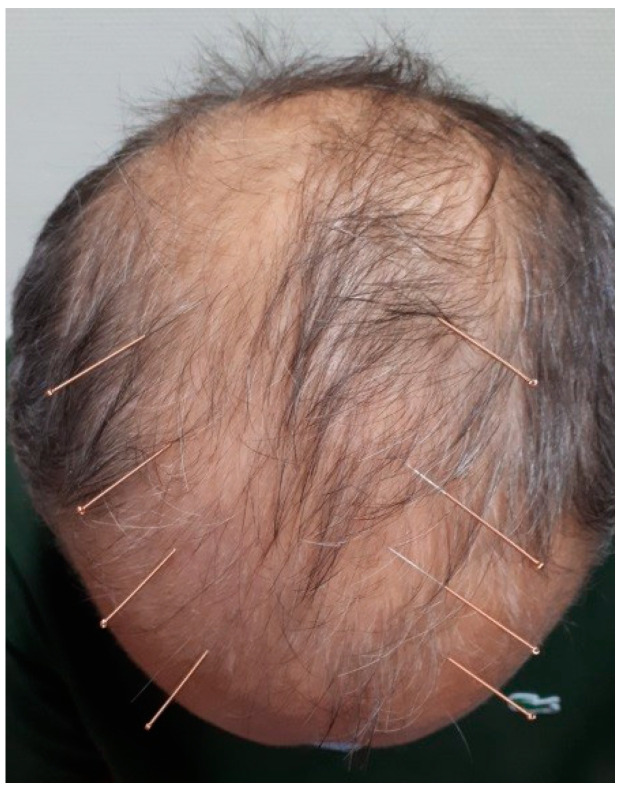
Schematic diagram of the puncture technique ˂15° and oblique angle puncture direction.

**Figure 2 medicines-09-00044-f002:**
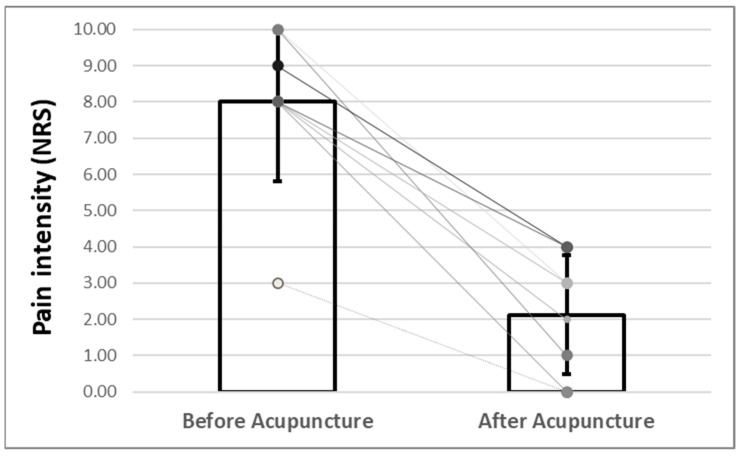
Results of pain intensity assessment before and after acupuncture treatments by therapy session. The pain intensity (expressed in NRS) sank after every single session, showing a maximum pain reduction of −9 points (from 10 to 1 NRS) in one single session.

**Figure 3 medicines-09-00044-f003:**
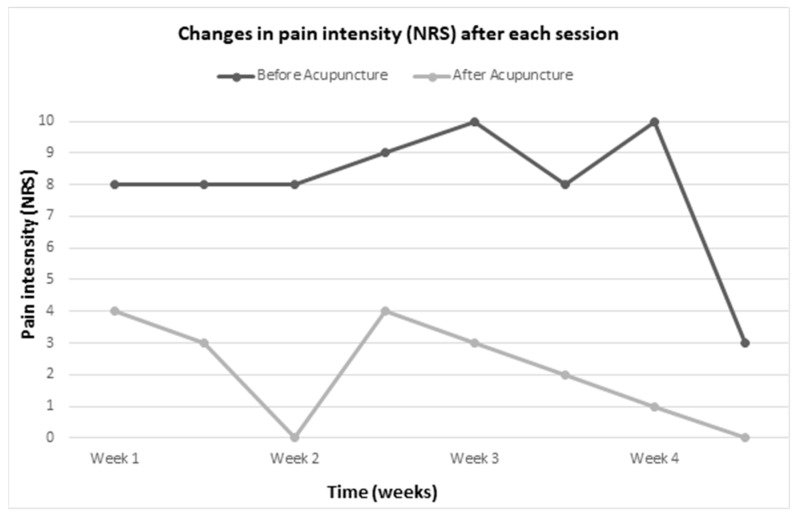
Time-lapse of changes in pain intensity (NRS) after each session.

**Table 1 medicines-09-00044-t001:** Legend: Daily medication and dosage.

Medication	Dosage
Clonazepam 0.5 mg	6×/day
Aspirin 100 mg	1×/day
Atorvastatin 20 mg	1×/day
Teriflunomid 14 mg	1×/day
Bisoprolol 2.5 mg	1×/day
Colecalciferol	2×/day
Candesartan plus 16/12.5 mg	1×/day
Duloxetine 60 g	1×/day
Gabapentin 700 mg	4×/day
Pantoprazole 40 mg	1×/day
Oxcarbazepine 600 mg	2×/day
Vitamin B-Complex	2×/week

**Table 2 medicines-09-00044-t002:** Results of pain intensity assessment using the NRS before and after acupuncture treatments by therapy session. m = arithmetic mean.

Session	Pre (m)	Post (m)	Difference (m)
1	8.0	4.0	4.0
2	8.0	3.0	5.0
3	8.0	0.0	8.0
4	9.0	4.0	5.0
5	10.0	3.0	7.0
6	8.0	2.0	6.0
7	10.0	1.0	9.0
8	3.0	0.0	3.0
**Total**	64.00	17.00	47.00

## Data Availability

All data used to support the findings of this study are included within the article.

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
