# Peer review of "Ashi Scalp Acupuncture in the Treatment of Secondary Trigeminal Neuralgia Induced by Multiple Sclerosis: A Case Report"

_medicines, 2022, doi:10.3390/medicines9080044_

Round 1

Reviewer 1 Report

I reviewed the case report manuscript entitled " Ashi Scalp Acupuncture In The Treatment Of Secondary Trigeminal Neuralgia Induced By Multiple Sclerosis: A Case Report". The authors tried to examine the efficiency of an acupuncture method in MS-related trigeminal neuralgia. It is a novel idea. However, there are some major concerns:

1- The abstract and introduction do not report sufficient evidence to support the aim of the study.

2- There are some English mistakes. 

3- Measurements of the effect of ASA are not enough. There should be further assessment, rather than only the pain severity. 

4- There are many details of the medications that are not needed to be charted in a separate table. 

5-  The follow-up is short, and it is hard to say ASA was a good procedure for the long term. 

6- It is better to use more questionnaires to assess the pain severity and its impacts on life. 

7- The discussion is too short, and a lot of evidences are not discussed. 

Author Response

1- The abstract and introduction do not report sufficient evidence to support the aim of the study.

Thank you for your comment. The abstract has accordingly been amended.

Acupuncture as a non-pharmacotherapy could be a promising complementary therapy for the treatment of TN. MS gradual damage of the nerves might affect the muscles’ function. This can lead to acute or paroxysmal pain in the form of spasms that might progress to the formation of myofascial trigger points also known in Traditional Chinese Medicine as Ashi point (AP). Localizing these AP through palpation and pain sensation feedback by patients with MS is an indicator of disease progression.

Introduction

Multiple sclerosis (MS) is an autoimmune-mediated disorder that affects the central nervous system leading to chronic illness affecting the brain and spinal cord leading to a neurological disorder. The underlying mechanisms of an MS attack is an aggressive inflammation process in areas of the white matter of the central nervous system in random patches called plaques. Upon the formation of these plaques, the destruction of myelin sheaths is triggered. Myelin is the insulating sheath of the nerves that promotes smooth and high-speed transmission of electrochemical messages between the brain, the spinal cord, and the rest of the body. The damage of the myelin sheath leads to disruptive neuronal communication and slowed electrochemical signaling. The blockade of neuronal transmission cause loss of function and somatosensory disturbance

2- There are some English mistakes.

Thank you for your comment. The manuscript has been proofread and corrected by a native English speaker. 

3- Measurements of the effect of ASA are not enough. There should be further assessment, rather than only the pain severity. 

Thank you for your comment. 

MS is a progressive disease, therefore the goal of our research was to establish whether or not the pain symptomatic was regressive after ASA. It is very common in our rehabilitation center to evaluate pain intensity assessed only by NRS on patients suffering from pain before and after acupuncture. Therefore we used the same method with the patient in the study. In addition a global impression of the improvement has been asked from the patients. However, this is not a validated scale.
An option for the future could be to additionally implement The West Haven-Yale Multidimensional Pain Inventory, before the acupuncture intervention and after the last intervention (3-4 weeks). 

4- There are many details of the medications that are not needed to be charted in a separate table. 

Thank you for your comment.

We believe that details concerning the medication play a very important role in order to get a complete overview of the patient’s situation. As recommended, we have simplified the table as follows: 

Medication                                                                              Dosage

Clonazepam 0.5mg                                                                6x/day

Aspirin 100mg                                                                        1x/day

Atorvastatin 20mg                                                                  1x/day

Teriflunomid 14mg                                                                 1x/day

Bisoprolol 2.5mg                                                                    1x/day

Colecalciferol                                                                          2x/day

Candesartan plus 16/12.5mg                                                  1x/day

Duloxetine 60g                                                                        1x/day

Gabapentin 700mg                                                                 4x/day

Pantoprazole 40mg                                                                1x/day

Oxcarbazepine 600mg                                                           2x/day

Vitamin B-Complex                                                               2x/week

5-  The follow-up is short, and it is hard to say ASA was a good procedure for the long term.

Thank you for your comment.

This is the limitation of this study. Due to the pandemic situation, we were not able to follow up the patient. Thus, we could achieve a short-term effect in pain reduction after each acupuncture session and intra-session applying ASA.   

6- It is better to use more questionnaires to assess the pain severity and its impacts on life. 

Thank you for your comment. We do agree.  

Our multimodal pain management comprises the following therapeutic elements: psychoeducation and behavioral therapy, physiotherapy, endurance training, relaxation therapies, manual therapies, as well as acupuncture. Had we used other assessment tools, we would not have been able to evaluate whether the pain reduction intra-session resulted from acupuncture or from a different therapy. 

We ask for your understanding, but the main outline measure was to quantify pain intensity using NRS before and after each acupuncture session. We only assessed NRS as a parameter to evaluate the pain and did not lay focus on the quality of life, as it was irrelevant to the scope of this case report.  

7- The discussion is too short, and a lot of evidences are not discussed. 

Thank you for your comment. We do agree.

In order to provide further evidence, we have revised our discussion and provided more references from previous reviews, meta-anaylses and clinical trials. These studies all show significant benefits provided by acupuncture, not only in MS patients, but also in a broad spectrum of neurological and non-neurological diseases.

Reviewer 2 Report

Dear Authors,

Your manuscript is really interesting and well conducted.

I have some minor revision for your manuscript.

1) Please explain better the MS history of the patients (which are onset symptoms?)

2) In the discussion section you mentioned that scalp acopuncutre can be found in other reports, could you discuss this part extensively ? Which is the typical patient that could benefit from this type of acupuncture?

Author Response

1) Please explain better the MS history of the patients (which are onset symptoms?)

Thank you for your comment.

We have added the following information to the patient’s history:

“He was diagnosed with MS in 2014. At that time, his main symptom was a recurrent pain (in the context of TN) in the right half of his face. As an in-patient in another RC in 2016, treatment was implemented for his repeated psychosomatic state of exhaustion, which had also been an early symptom.”

2) In the discussion section you mentioned that scalp acupuncture can be found in other reports, could you discuss this part extensively? Which is the typical patient that could benefit from this type of acupuncture?

Thank you for your comment. We have discussed the part extensively and provide information about the patients, who would benefit from scalp acupuncture.

In literature, many reports about scalp acupuncture (SA) can be found. Hao et al. (2013) concluded in their case report that SA was a better therapy approach in bringing about rapid advancement in patients with MS correlated to other acupuncture methods to the same degree as acupuncture on the ear, body, and hand [17].  The initial SA interference after acute ischaemic stroke with hemiplegic paralysis showed in a randomized controlled clinical trial the enhanced myodynamia of the afflicted limbs, improvement of neurological deficiency severity and quality of life [18]. A meta-analysis result reported that SA improved motor function in patients with stroke whether ischemic or hemorrhagic [19]. In a systematic review and meta-analysis evaluating the efficacy and safety of SA for insomnia the authors could provide insight of the effectivity and safety treating insomnia [20]. Liu et al. (2019) conducted a systematic review and meta-analysis about SA for children's autism spectrum disorders (ASD).  The main results in their meta-analysis advocated that in comparison with behavioral and educational interventions, SA was more effective in lowering Childhood Autism Rating Scale and Autism Behavior Checklist scores and promoting Psychoeducational Profile scores in communication, physical ability, and behavior. Accordingly, the authors ensured SA may have positive effects on the treatment for children affected by ASD [21]. Therefore, patients with MS, affected by stroke, children with autism spectrum disorders, suffering from insomnia might benefit from AS.

Round 2

Reviewer 1 Report

The manuscript is improved, however, I am still unsure about the acceptance. My main concern is the incomplete pain assessment and short follow-up of the patient, which makes the uncertainty of conclusion.